# Study on the Effect of Contrast Agent on Biofilms and Their Visualization in Porous Substrate Using X-ray μCT

**Abhishek Shastry [1,2,\*]**, **Xabier Villanueva [3]**, **Hans Steenackers [3]**, **Veerle Cnudde [2,4,5]**, **Eric Robles [6]** and **Matthieu N. Boone [1,2,\*]**

1    Radiation Physics Research Group, Department of Physics and Astronomy, Ghent University, Proeftuinstraat 86/N12, B-9000 Gent, Belgium
2    Centre for X-ray Tomography, Ghent University, Proeftuinstraat 86/N12, B-9000 Gent, Belgium; Veerle.Cnudde@UGent.be
3    Microbial Communities & Antimicrobials (MICA) Laboratory - www.micalab.be, KU Leuven, Centre of Microbial and Plant Genetics, Kasteelpark Arenberg 20, B-3000 Leuven, Belgium; xabier.villanueva@kuleuven.be (X.V.); hans.steenackers@kuleuven.be (H.S.)
4    Pore-Scale Processes in Geomaterials Research Group (PProGRess), Department of Geology, Ghent University, Krijgslaan 281/S8, B-9000 Gent, Belgium
5    Environmental Hydrogeology, Department of Earth Sciences, Utrecht University, Princetonlaan 8a, 3584 CB Utrecht, The Netherlands
6    The Procter and Gamble Company, Newcastle Innovation Center, Whitley Road, Longbenton, Newcastle-Upon-Tyne NE12 9TS, UK; robles.es@pg.com
\*    Correspondence: shastryabhishek91@gmail.com (A.S.); Matthieu.Boone@UGent.be (M.N.B.)

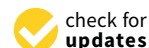

**Featured Application: The proposed method may be adopted for in-situ visualisation of low virulence gram-negative bacterial strains using contrast-enhanced X-ray μCT. The use of different techniques in this study enables future researchers to evaluate the toxicity of other contrast agents on biofilms.**

**Abstract:** Investigation of biofilms and visualization using non-destructive imaging techniques like X-ray μCT has recently gained interest. Biofilms are congregations of microorganisms that attach to surfaces and comprise of microbial cells embedded in extracellular polymeric substances (EPS). They are ubiquitous entities that are commonly found in any non-sterile setting and have direct implications on human health. Methods to visualize them in situ are highly needed to understand their behaviour (attachment and detachment) inside a substrate. Contrast-enhanced X-ray μCT is a 3D imaging technique that is capable of visualising objects that have very low attenuation contrast. The use of contrast agents in X-ray μCT has been an evolving process, however, the possible toxic effect of these chemical compounds against biofilms has not been studied in detail. In this study, we focus on the toxic effect of contrast agents and study the diffusion and drainage of contrast agents in biofilms. We propose using water-soluble potassium bromide (KBr) as a suitable contrast agent for enhancement of the attenuation coefficient of a monoculture of *Pseudomonas fluorescens* biofilms inside a porous substrate. At the given concentration, KBr proved to be less bactericidal compared to other commonly used contrast agents and at 5% *w/v* concentration we were able to clearly distinguish between the biofilm and the porous substrate.

**Keywords:** X-ray μCT; biofilms; contrast agents; in-situ; flow cell

---

## 1. Introduction

Bacteria do not live in isolation, but in bacterial communities [1,2] in which they can have several kinds of interactions, including cooperation and competition [3]. One notorious form of bacterial communities are biofilms [4], which can be defined as bacterial aggregates located on an interface and surrounded by a protective layer of extracellular polymeric substances (EPS) [5]. These biofilms offer several benefits to the bacteria, including increased attachment to surfaces and protection against external stresses, which include biocides and antibiotics [6]. The EPS in the biofilm forms a complex mesh of polysaccharides, extracellular DNA (eDNA) and proteins around the bacteria, resulting in decreased penetration of external substances [4]. This decreased penetration of external substances also results in oxygen and nutrient gradients inside the biofilm, generating subpopulations of bacteria with decreased metabolic activity that are less sensitive to antimicrobial compounds [7].

Biofilms are commonly found in natural, medical, and industrial settings and have direct implications on humans [8]. In certain cases, bacterial biofilms can be beneficial and important in biotechnological applications (for production of bulk chemicals and enzymes) [9,10] and can be applied in agriculture (bioremediation and growth promotion) [11], wastewater treatment [12] and as an anticorrosive strategy [13]. However, given their increased tolerance to disinfectants (glutaraldehyde, sodium hypochlorite) [14], antimicrobial agents [15], metal ions and halogenated ions [16–18], biofilms [16,19] also often result in persistent infections and can be a source of several problems, like biofouling, pipeline clogging and product contamination [20]. For these reasons, biofilm control and detection are important.

Biofilms are dependent on the environmental conditions and are unique to the bacterial strains that produce them. *Pseudomonas fluorescens* Pf01 is a gram-negative aerobic bacterium, which is capable of developing biofilms under stress conditions. Because of its low virulence [21], it is an ideal model organism to study biofilm formation, since it can be used in facilities that do not contain a biosafety containment facility.

Porous substrates can be defined as substrates that consist of a solid matrix and pores, which allow for external matter to pass through. Because porous substrates are universal, they are commonly encountered in our day to day life. Some of them include household cleaning implements like kitchen sponges and wipes. The fluid flow inside these substrates is not only influenced by the fluid properties like density, viscosity and capillary pressure but also depends on the pore geometry and material of the substrate. Due to their complex porous structure, they are prone to the imbibing of unwanted residual soils which may lead to the development of biofilms overtime. Hence there is a need to understand the development of biofilms inside the porous substrates and visualize their distribution non-invasively.

Commonly used techniques to visualize biofilms are not suited for imaging porous materials. Optical microscopy [22] and fluorescence microscopy [23] are used extensively for visualisation of biofilms. However, these techniques are limited to 2D visualisation of the sample and involve extensive sample preparation. In the case of fluorescence microscopes, there is a need to use expensive fluorescent stains. Other techniques like Transmission electron microscopy (TEM) [24] and confocal laser scanning microscopy (CLSM) [25] are powerful tools for the characterisation of biofilms and provide a 3D representation of the specimen. However, in the case of TEM, there is a need for fixation of the sample, followed by dehydration and a polymerisation process with epoxy resin. This extensive treatment may lead to considerable distortion of the sample [26]. Likewise, for CLSM the depth of penetration for imaging is limited and it is difficult to visualize biofilms in-situ in a thick porous substrate. These restrictions highlight the need for 3D imaging techniques like X-ray µCT that allow the user to visualize the internal and external structure of objects [27] non-invasively and non-destructively. X-ray µCT is one of the well-established techniques to study different porous substrates and visualize 3D geometry [28,29]. X-ray µCT data can be used as an input for pore-scale modelling [30] and flow simulations [31] to predict the flow and unwanted residual development inside the porous substrate. X-ray µCT is also being used to study the biofilm morphology and its permeability properties in porous substrates [32,33].

The low density of the EPS components, such as polysaccharides and a variety of proteins, glycoproteins, and glycolipids, however, makes it difficult to visualize biofilms using X-ray μCT. Although the application of chemical compounds for contrast enhancement between the biofilms and the substrate has been in place for some time, no ideal contrast agent has thus far been identified, taking into account the boundary conditions of minimal impact on the biofilm properties and growth, while providing sufficient and homogeneously distributed contrast. Davit et al. [34] used the idea of applying two chemical compounds, barium sulphate and potassium iodide ($BaSO_4$ and $KI$) as contrast agents. A medical suspension of $BaSO_4$ was used to differentiate between the aqueous phase (growth medium) and the biofilm, while $KI$ diffused inside the biofilm and aided in delineation between the biofilm and porous medium. However, since $BaSO_4$ is not water-soluble, it interacts with biofilms and displaces them. Moreover, the sedimentation of $BaSO_4$ results in a non-homogenous distribution inside the substrate. In the experiment by Carrel et al. [35], iron sulphate ($FeSO_4$) was used as a contrast agent and was supplemented during the growth of bacteria. In their observation, they found that the biofilm grown in the flow cell exhibits brown patches due to the presence of iron oxide. Therefore, the colloidal iron formed due to oxidation of Fe (III) may lead to irreversible chromism effects on the porous substrate. Similarly, 1-chloronaphthalene was used as a contrast agent by Smułek et al. [36]. However, 1-chloronaphtalene is known to be highly toxic, which was also observed during our tests to reproduce these results, as it chemically reacts with the substrate resulting in its dissolution.

Given the aforementioned limitations, an alternative chemical compound with different physical and chemical properties, namely potassium bromide ($KBr$), was evaluated here to stain the biofilm *P. fluorescens* Pf01. Aqueous solutions of $KBr$ are neutral (pH 7) and do not require any addition of buffer and hence can be applied onto the mature biofilms easily. Rendleman [37] has compared various cations for their affinity and stability towards aggregative polysaccharides like alginic acid. His study showed that alginic acid has higher affinity towards $K^+$ cations than other ions in the same periodic group. As the *P. fluorescens* bacteria has the potential to synthesize alginic acid [38], the $K^+$ cations attach to alginic acid while Br readily binds to the nucleic acid [39] present in the biofilms. The linear X-ray attenuation coefficient is dependent on the chemical composition and density of the sample. It is directly proportional to the atomic number of the chemical element under observation. As $KBr$ has higher density and both the elements K and Br have higher atomic numbers compared to CHON (carbon, hydrogen, oxygen, nitrogen) elements present in the biofilm and the surrounding medium, they exhibit higher X-ray attenuation at given X-ray energy. Comparatively, Br with higher atomic may contribute more in enhancing the contrast.

## 2. Materials and Methods

### 2.1. Growth Medium for Biofilm Culture

*Pseudomonas fluorescens* Pf01 bacteria come under the category of environmental bacteria and are commonly present in soils, plants and water. They grow ideally between 25 °C and 30 °C and produce biofilms [40]. Overnight (ON) cultures of *P. fluorescence* were prepared at 30 °C in Lysogeny broth (LB) medium according to the standard protocol formulated by Luria et al. [41]. These cultures were diluted 1/100 in Tryptic Soy Broth (TSB, Becton Dickinson GmbH, Germany) 1/10 to obtain an inoculation density of $10^6$ CFU/ml and incubated at 28 °C in the biofilm assays. This set-up was employed to maximize biofilm formation as the previous data showed that bacteria develop more biofilm using diluted media. Phosphate-buffered saline (PBS) was prepared using standard protocol commonly used in microbiology [42]. As the biofilm grew predominantly at the liquid-air interface, after the drainage of TSB, PBS was used to remove the unattached biofilms from the substrate and also to remove the excess of contrast agent solution applied for contrast enhancement.

## 2.2. Porous Substrate and Flow Cell

For the incubation of bacteria, a model porous substrate close to real-case was fabricated using polymethylmethacrylate (PMMA). The flow cell in Figure 1 consists of a cylindrical tube with a grid plate located in the middle of the tube. The latter acts as an experimental porous substrate to hold the biofilm during the drainage of the growth medium from the tube. The tube has a provision for an inlet (lid) and outlet, facilitating the replacement of the growth medium. The height of the entire tube is 5 cm with an inner diameter of 1.4 cm. The wall thickness of the tube is 0.1 cm. The growth medium and bacterial inoculum are added through the lid using micropipette. The outer diameter of the lid has a tolerance of 0.05 cm enabling it to have a clearance fit with the tube. The grid plate with a thickness of 0.5 cm consists of 7 holes of 0.15 cm diameter, built in accordance with the average pore diameter of a commercially available cellulose kitchen sponge [43]. The outlet is made at the central axis of the tube below the grid with a diameter of 0.2 cm. The flow through the outlet is managed with the help of a screw. PMMA with its low X-ray attenuation ensures that sufficient X-ray flux reaches the detector, thereby justifying its selection for the construction of the flow cell.

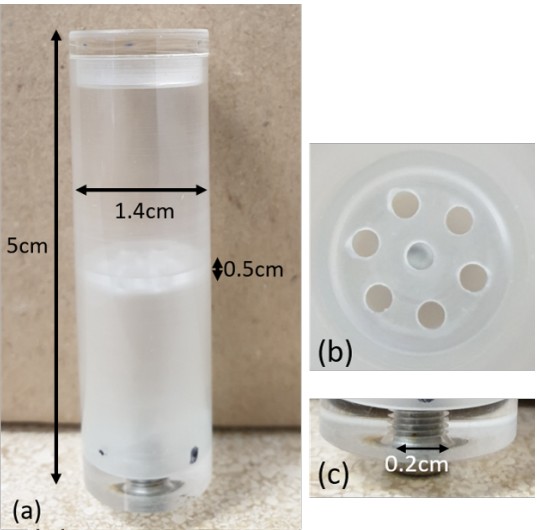

**Figure 1.** (**a**) Photo of the flow cell. (**b**) Grid plate with 7 holes of 0.15 cm diameter (**c**) Outlet of the flow cell with a screw to regulate the flow.

## 2.3. Contrast Agents

For the experiments, Phosphotungstic acid (PTA), potassium iodide (KI), zinc sulphate (ZnSO$_4$), iron sulphate (FeSO$_4$), and potassium bromide (KBr) were selected (VWR International, Geldenaaksebaan, Oud-Heverlee, Belgium). The three experiments were conducted to evaluate the use of these chemicals as a contrast agent for biofilms visualization. As a first experiment, each of these chemical compounds were dissolved in the growth medium at a working concentration of 2% $w/v$ and the bacteria were incubated to develop biofilms. In the second experiment, the KBr concentration in the growth medium was raised to 5% $w/v$ and bacteria were incubated in the contrasted growth medium. The third experiment involved the administration of 5% $w/v$ KBr solution to the fully developed biofilms. All the results of these experiments were compared with the control batch to determine the toxic effect of each of the contrast agents.

## 2.4. Scanner System and Microscopy

In this work, we used an X-ray µCT system custom-designed by the Ghent University Centre for X-ray Tomography (UGCT). The Environmental Micro-CT system or EMCT [44] is a rotating gantry-based system (medical CT principle) specifically designed to conduct in-situ 4D-µCT

experiments. In this system the source-detector rotates around the stationary object, thus allowing us to have provision for fluid flow in the in-situ device. With an L9181-02 X-ray tube (Hamamatsu Photonics, Hamamatsu, Japan) and a Xineos 1313 flat-panel detector (Teledyne DALSA, Waterloo, ON, Canada), this X-ray μCT system is optimized to acquire fast dynamic scans. For more information about this X-ray μCT system and the relationship between CT scanning speed and resolution, the reader is referred to the article by Dierick et al. [44]. Alternately, other micro-CT systems can be used to visualize biofilms at the given scanning conditions elaborated in this paper. However, it is difficult to conduct the diffusion experiment with rotating-object CT systems as the rotational movement of the object is likely to induce motion artifacts. Reconstruction of the radiographs obtained during X-ray μCT scans was done using Octopus Reconstruction [45] which is an in-house developed software package. Further, Octopus Analysis [46] was used for the 3D analysis of the reconstructed images to know the change in the attenuation coefficient value of biofilms over time and depth. To image the morphology of bacteria incubated in the growth medium with and without contrast agent, Zeiss LSM 880 confocal laser scanning microscope (Carl Zeiss Microscopy GmbH, Jena, Germany) was used. For staining bacteria, LIVE/DEAD BacLight Bacterial Viability kit (Life Technologies Europe BV—Invitrogen Division, thermofisher scientific) was used according to the manufacturer recommendations. The statistical comparisons of the toxicity of different contrast agents were made using one-way ANOVA (analysis of variance) with Dunnett post-hoc test. The differences were considered significant if the probability value ($p$-value) was below 0.05.

### 2.4.1. X-ray μCT Scanner Settings

For all the X-ray μCT experiments listed in Section 2.7, 1401 projections of 0.1 s exposure time per projection with a voxel size of $12 \times 12 \times 12$ μm$^3$ were acquired over the full 360° rotation. The tube output was set at 40 kV and 8 W and the duration of each scan was around 3 min.

### 2.5. Simulations of X-ray CT Scan

To find the minimal concentration of contrast agent that must be present in the substrate, virtual CT scans were simulated using Arion [47]. Using this tool, virtual CT scans for different concentrations of the contrast agent in water medium were simulated. An image representing the central slice of a similar flow cell containing grid holes are shown in Figure 2. Material properties (density and attenuation coefficient) were assigned to each of the constituents in the loaded image and the concentration of different contrast agents was varied. The grid holes assigned with a specific concentration of contrast agent were maintained in pure water because water constitutes nearly 90% of a biofilm [48]. Hence grid holes with contrast agent solution can be interpreted as contrasted biofilms. For all the phatoms (flow cell containing grid holes) generated at different concentration of contrast agent, simulated data were acquired using the same scanner settings as in the experimental data presented in Section 2.4.1.

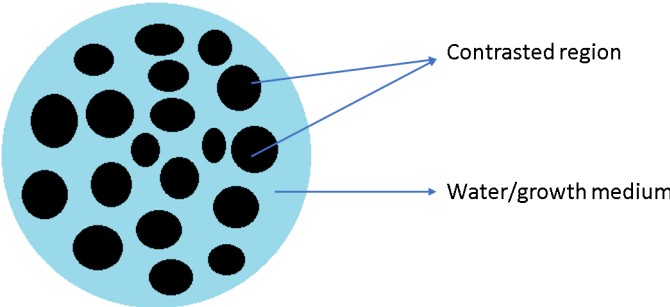

**Figure 2.** Representation of the grid plate with growth medium and contrast agent solution.

### 2.6. Determination of Contrast Agent's Toxicity

#### 2.6.1. By Plate Counting Method

To test the toxicity of several contrast agents listed in Section 2.3, *P. fluorescens* biofilms were grown in a small petri dish batch system in three biological repeats. *P. fluorescence* ON cultures were diluted 1/100 in TSB 1/10 containing different contrast agents at 2% *w/v* and incubated for 48 h at 28 °C. Simultaneously a control batch of bacteria (growth medium without contrast agent) was grown. Upon incubation, the planktonic phase was removed and replaced by sterile phosphate buffer saline (PBS) (pH 7.4) to remove all the loosely attached bacteria. Then, the washing PBS solution was removed, and 1 mL of fresh sterile PBS was added. The biofilms were recovered in 1.5 mL microcentrifuge tubes using a cell scraper and a 1 mL micropipette. After this step, the biofilms were homogenized by passing through a 1 mL syringe with a 25G needle, vortexed, and serially diluted. The dilutions were then plated on LB agar plates and incubated for 24 h. Finally, to quantify the bacterial population, the plate counting method [49] was used. Further, to understand the effect on the bacteria at higher concentration of KBr, they were incubated in 5% *w/v* KBr growth medium and the bacterial colonies were counted.

#### 2.6.2. By Confocal Laser Scanning Microscope

In addition to cell scraping and plate counting of *P. fluorescens* biofilms, a subset of bacteria was visualized by microscopy to determine the morphological changes in developed biofilms. Here 3 batches of bacteria were incubated for 48 h at 28 °C. While the first batch represented the control set (without application of 5% *w/v* KBr), the second batch included bacteria incubated in growth medium containing KBr (5% *w/v*). The third batch was used for acute toxicity tests wherein the incubated bacteria (48 h) was treated with 5% *w/v* of KBr for approximately 4 h. At the end of incubation time, all the three batches of developed biofilms were imaged at 40× magnification using a Zeiss LSM 880 confocal laser scanning microscope (Carl Zeiss Microscopy GmbH, Jena, Germany).

### 2.7. Protocol for Bacterial Incubation in Flow Cell

An ON culture of *P. fluorescens* was diluted 1/100 in the flow cell containing TSB 1/10. This setup was incubated at 28 °C for 48 h, for the bacteria to proliferate and develop biofilms. After 48 h the TSB 1/10 in the flow cell was replaced with fresh TSB 1/10 and further incubated for 24 h. This step helped in achieving more biomass on the grid plate inside the flow cell [50]. This protocol was followed for all the experiments involving bacterial incubation in the flow cell.

### 2.8. Determination of Irradiation Effect of X-rays on Biofilms

For this experiment, the bacteria were incubated in six flow cells simultaneously following the same procedure as elaborated in Section 2.7. After 72 h of incubation, the growth medium was drained out carefully and was replaced with 5% *w/v* KBr solution. The 5% *w/v* KBr solution was allowed to interact with the biofilm for half an hour. Three out of 6 flow cells were placed on the EMCT stage and were irradiated, with X-rays using the scanner settings mentioned in Section 2.4.1, for 6 min (in accordance with total experimental scan time as discussed in Section 2.10) while the remaining 3 flow cells acted as a control (without any exposure to X-rays). After this step, the contrast agent was removed from all the flow cells and PBS was flushed through the tube. Later, 1 mL of PBS was added and all the six flow cells were vortexed to detach biofilms from the surface of the tube. The solutions from each of the 6 flow cells were then transferred to 1.5 mL microcentrifuge tubes and were serially diluted. The dilutions were then plated on LB agar plates and incubated for 24 h. Afterwards, the bacterial population was determined by plate counting.

### 2.9. Diffusion Experiment

X-ray μCT was used to analyse the diffusion process of the contrast agent in biofilms. Following the protocol in Section 2.7, the bacteria were incubated for 72 h and then applied with 5% *w/v* KBr contrast agent solution. This setup was scanned immediately using the scanner settings described in Section 2.4.1 to record the diffusion of contrast agent into biofilms present in the grid holes. After 1 h of application of contrast agent solution, another X-ray μCT scan of the setup was acquired. Subsequently, the contrast agent solution was drained out carefully and PBS was applied to the contrasted biofilms. Again, two X-ray μCT scans were acquired: one immediately after submersion and one after 1 h of diffusion.

To determine the change in the attenuation coefficient of biofilms developed in the grid holes, two of the image slices from the initial and final scan at the same position of the flow cell were chosen. Comparison of these two values gave an estimate of the overall enhancement in the attenuation coefficient value of biofilms due to application of contrast agent over time. To analyse the diffusion rate, Octopus Analysis was used. The Volume of Interest (VOI) was determined by a single threshold on the data set obtained in Section 2.9, which had the highest contrast, and used for all other scans. The holes were labelled as indicated in Figure 3a. A graph was plotted along the length of the hole to determine the change in the attenuation coefficient of biofilms present inside the grid holes, over time. The rectangular block as shown in Figure 3b representing the stack of the images was chosen in such a way that the diffusion of contrast agent into biofilms developed on the grid plate is not missed. The graphs in Section 3.4 were plotted along the length of the pores over time.

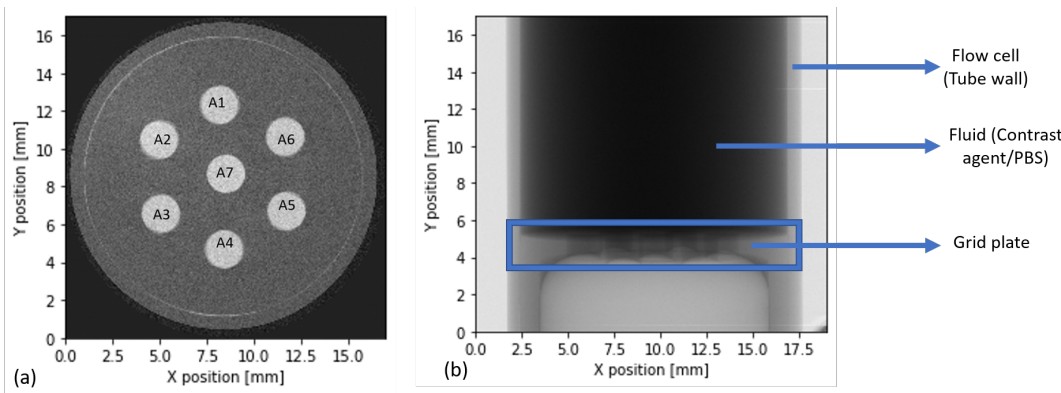

**Figure 3.** (**a**) Image slice with the labelling of pores. (**b**) Radiograph of the tube with a grid plate with the application of contrast agent/PBS. The rectangular block represents the chosen stack of images featuring the grid plate for analysis.

### 2.10. In-Situ Experiments for Biofilm Visualization

This experiment was aimed at visualizing biofilms developed in the flow cell using X-ray μCT. Initially, the flow cell was scanned using X-ray μCT in the absence of growth medium to capture the porous substrate before the incubation of bacteria. After the initial scan, the flow cell was placed on the CT stage and the 72-h protocol for bacterial incubation (Section 2.7) was followed. Afterwards, the TSB was carefully drained out making sure the biofilms formed at the interface between the TSB and air settled onto the grid plate. Then, 5% *w/v* of KBr solution was applied to the system and the contrast agent was allowed to diffuse into the fully grown biofilms for half an hour. After this step, the contrast agent solution was drained out and the PBS was flushed to remove any unwanted contrast agent that did not attach to the biofilms. X-ray μCT scans were acquired at the end of each step.

Characterisation of Biofilm

The 2D cross-sectional images of the grid plate obtained after drainage of TSB (before application of contrast agent) and PBS wash (after application of contrast agent) at the same position were subtracted to obtain the contrasted biofilms.

The following Table 1 summarizes the incubation protocols of different experiments mentioned in Sections 2.8–2.10 conducted in this study.

**Table 1.** Overview of all the experiments and their respective protocols for incubation of bacteria in a flow cell. (PBS application* step represents the addition of PBS for diffusion and PBS wash** is the passage of PBS through the grid plate after contrast agent application).

| List of Experiments | ON Culture of Bacteria at 30 °C | Incubation of Bacteria in TSB 1/10 for 48 h | Replacement of TSB 1/10 and Further Incubation for 24 h | Application of 5% *w/v* of KBr to Fully Developed Biofilms | PBS Application* | PBS Wash** | Serial Dilution and Plate Counting |
|---|---|---|---|---|---|---|---|
| Irradiation effect of X-rays on biofilms | ✓ | ✓ | ✓ | ✓ | | ✓ | ✓ |
| Diffusion experiment | ✓ | ✓ | ✓ | ✓ | ✓ | | |
| In-situ Experiment for biofilm visualization | ✓ | ✓ | ✓ | ✓ | | ✓ | |

## 3. Results

### 3.1. Simulated Data of the Virtual Sample with KBr

Two virtual samples (Figure 2) with grid holes representing the contrasted biofilms at 2% *w/v* and 5% *w/v* KBr were simulated using Arion. The attenuation coefficient values of these two simulated contrasted biofilms were determined using Octopus Analysis (Section 2.9). In Figure 4, we can see that the attenuation coefficient values of 2% *w/v* KBr (a) and 5% *w/v* KBr (b) in the grid holes are 1.17 cm$^{-1}$ (standard deviation is 0.071) and 2.4 cm$^{-1}$ (standard deviation is 0.12), respectively, and the average attenuation coefficient of the surrounding water medium is around 0.45 cm$^{-1}$ (in correlation with the real value with standard deviation 0.03).

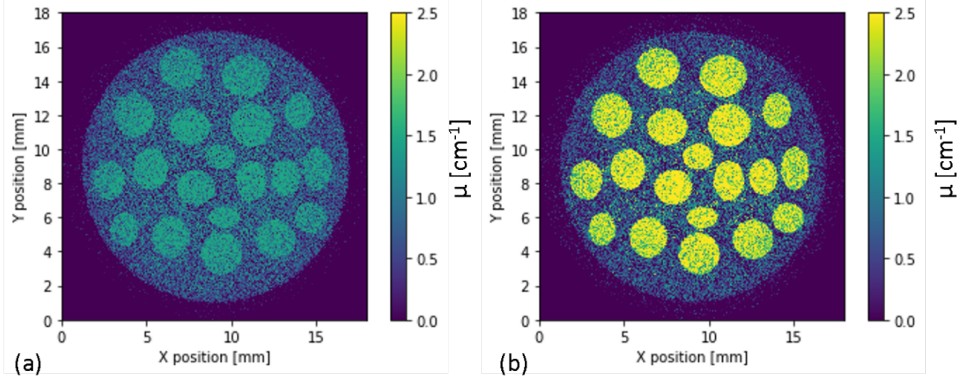

(a)　　　　　　　　　　　　　　(b)

**Figure 4.** (**a**) Reconstructed image of the grid plate with water and contrast agent (2% *w/v* KBr solution). (**b**) Reconstructed image of the grid plate with water and contrast agent (5% *w/v* KBr solution). The bright yellow spots represent the contrasted grid holes filled with 5% *w/v* of KBr solution.

Here, as an ideal case, for simulation, it was considered that the concentration of contrast agent inside the biofilms and the concentration of the initially applied KBr solution are the same.

However, it should be noted that the concentration of contrast agent inside the biofilm in real in-situ experiments is unknown and the scope to control it is limited. Therefore, despite the attenuation coefficient of 2% $w/v$ KBr being 2.5 times more than the attenuation coefficient of water, slightly higher concentration i.e., 5% $w/v$ KBr was used in the in situ experiments to enhance the contrast gradient between the constituents in the flow cell. A comparison is drawn between the attenuation coefficient values of the biofilm present in the grid hole for the simulated and the real experiments in the discussion section.

### 3.2. Long-Term Toxicity Test of Contrast Agents on P. Fluorescens Pf-01 Biofilms

Figure 5 depicts the effect of different contrast agents on the population size of biofilms compared to control batch. In the case of 2% $w/v$ FeSO$_4$ and ZnSO$_4$, the bacterial population size was significantly reduced ($p = 0.0023$ and $p = 0.0023$, respectively) by around 5 $log_{10}$ and 4 $log_{10}$, respectively. Similarly, for 2% $w/v$ PTA and KI there was a significant impact ($p = 0.0034$ and $p = 0.0112$, respectively) on the population size with a reduction by 1 $log_{10}$ and slightly less than 1 $log_{10}$, respectively. Therefore, no further experiments were conducted using these chemical compounds as contrast agents. At 2% $w/v$ KBr there was no significant ($p \geq 0.05$) reduction in the bacterial population size. However, at 5% $w/v$ KBr the bacterial population size was reduced significantly ($p = 0.0031$) by 1 $log_{10}$.

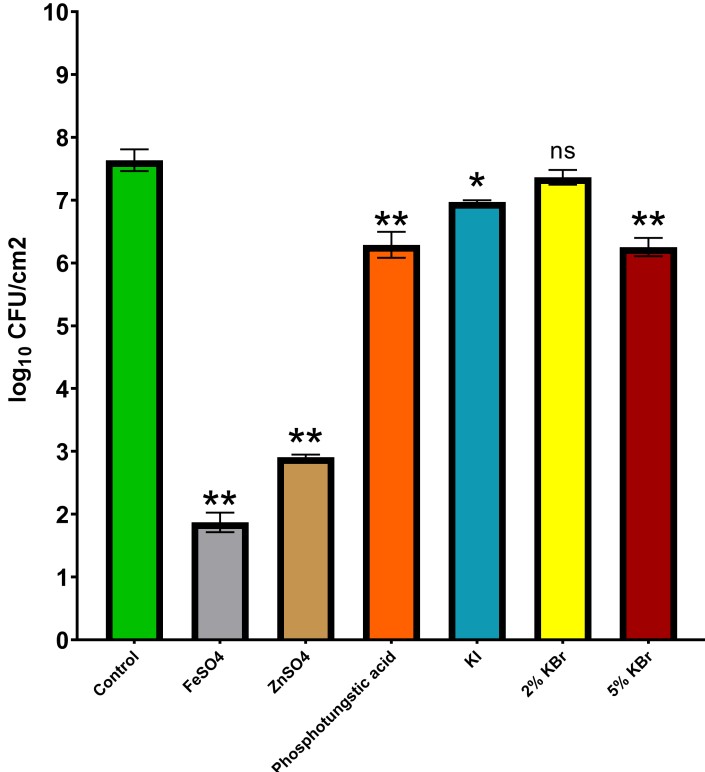

**Figure 5.** Individual population size of biofilms incubated in each of the contrast agent solutions for 48 h at 28 °C. Data are represented as the mean with SEM (Standard Error of Mean). Statistical comparisons were made using analysis of variance (ANOVA). In the figure, * and ** indicates a significant difference $p < 0.05$ and $p < 0.01$, respectively. While ns represents no significant difference ($p \geq 0.05$) when compared to control.

Next to its effect on the number of biofilm cells, the influence of 5% $w/v$ KBr on the bacterial biomass was assessed by crystal violet staining. The biofilm biomass per cell was found to be increased (Table 2) probably due to the stress induced by the presence of KBr in the growth medium.

**Table 2.** Measured total biomass and biomass/cell of *P. fluorescens* Pf-01 bacteria in control and 5% *w/v* KBr solution. (std—standard deviation).

| Incubation Type | CFU/cm$^2$ | Total Biomass (OD) | std | Biomass/Cell | std |
|---|---|---|---|---|---|
| Control | $3.62 \times 10^7$ | 0.13 | 0.01 | $1.33 \times 10^{-10}$ | $2.03 \times 10^{-11}$ |
| 5% KBr | $2.25 \times 10^6$ | 0.10 | 0.02 | $7.78 \times 10^{-9}$ | $6.02 \times 10^{-9}$ |

In addition, the effect of 5% *w/v* KBr was investigated by confocal microscopy combined with Live/Death staining. Figure 6 indicates that the presence of 5% *w/v* KBr affects the phenotype of the bacteria, even if it does not impede the bacterial growth (bacteria still alive). There was cell elongation [51] observed after long term exposure to 5% *w/v* KBr suggesting an envelope stress response by the bacteria [52].

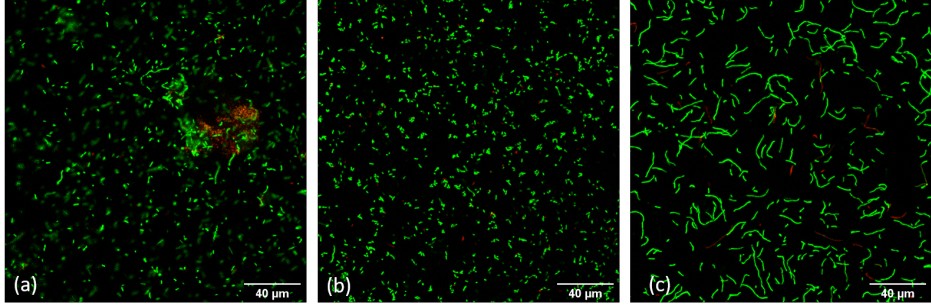

**Figure 6.** Comparison between bacterial biofilms grown in: (**a**) absence of KBr 5% *w/v* for 48 h, (**b**) absence of KBr 5% *w/v* for 44 h and subsequent addition of KBr 5% *w/v* for 4 h, and (**c**) presence of KBr 5% *w/v* for 48 h at 28 °C. The Live/Dead images were obtained at 40× magnification using a Zeiss LSM 880 confocal laser scanning microscope.

### 3.3. Irradiation Effect of X-rays on P. Fluorescens Pf-01 Biofilms

In Figure 7 we can see that the short-term irradiation during X-ray μCT scanning in the presence of contrast agent does not affect the viability of bacterial biofilms. There is no significant ($p \geq 0.05$) change in the population size of biofilms before and after irradiation of X-rays.

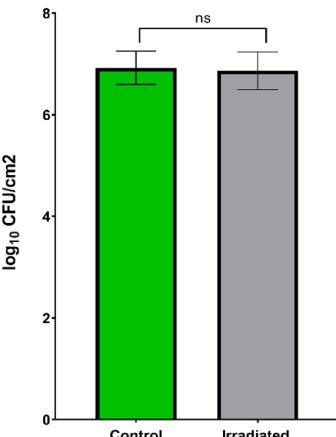

**Figure 7.** The number of colony-forming units (CFU) per cm$^2$ of the petri dish for control biofilm and CT scanned batch. Data are represented as the mean with SEM (Standard Error of Mean). Statistical comparison was made using analysis of variance (ANOVA). The ns represents no significant difference ($p \geq 0.05$.) when compared to control.

### 3.4. Diffusion Experiments

### 3.4.1. Contrast Agent Application

Figure 8 shows a small increment in the attenuation coefficient value of the pores after 1 h of application of contrast agent as compared to the scan acquired immediately after submersion with the contrast agent. This small change in the attenuation coefficient value over time suggests that the diffusion of contrast agent into biofilms occurs instantaneously upon exposure. Therefore, the application time was reduced to half an hour for the subsequent experiments. Although there is good penetration of the contrast agent along with the thickness of biofilms, even after 1 h of application equilibrium was not reached between the contrasted and non-contrasted part.

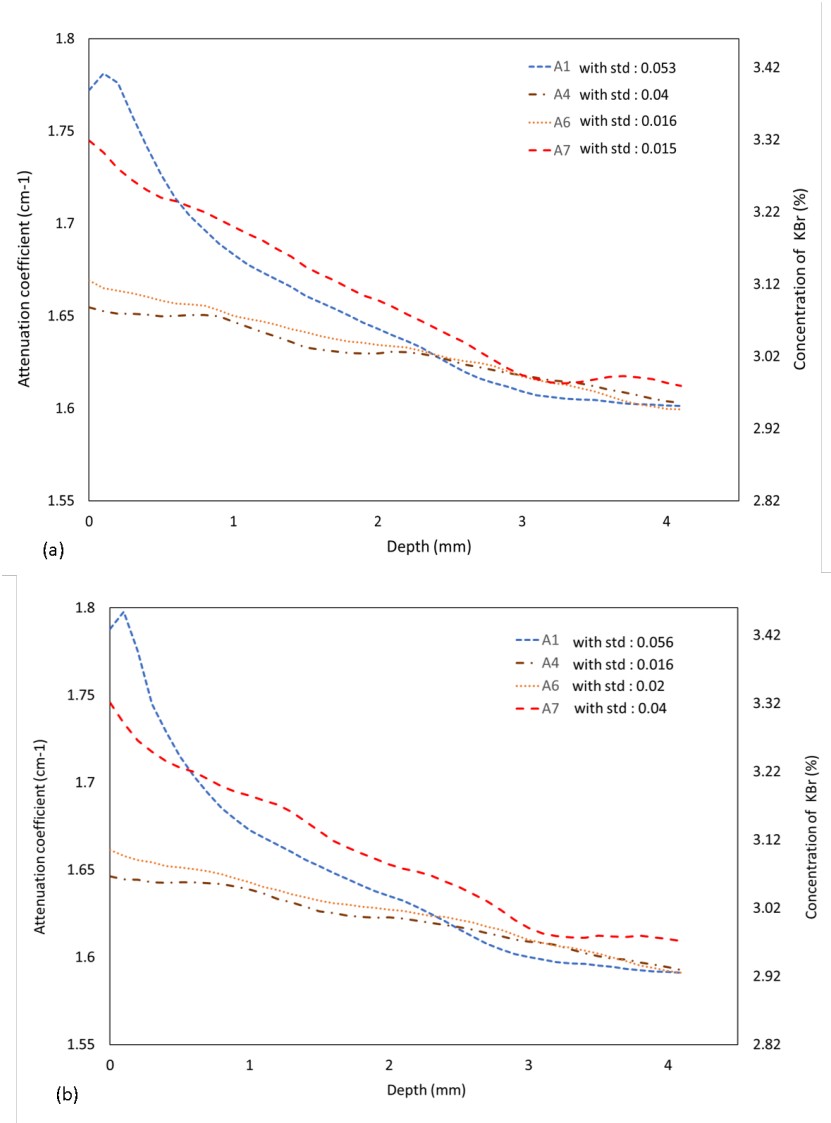

**Figure 8.** The measured attenuation coefficient in different pores as a function of depth due to the addition of contrast agent on biofilms (**a**) immediately acquired before the scan (**b**) after 1 h of application of contrast agent. The legend represents the grid holes as labelled in Figure 3a. (std—standard deviation).

### 3.4.2. Phosphate Buffer Saline Application

On the contrary, in Figure 9 for PBS application there is a significant change in the attenuation coefficient value of the pores over time. The flattening of the attenuation coefficient curve along the

depth of the pore suggests that the leaching out of the contrast agent from biofilms to PBS is uniform. But to attain the same attenuation coefficient value at all the pores, it takes about one hour, giving enough time to conduct X-ray µCT scan of the contrasted biofilms.

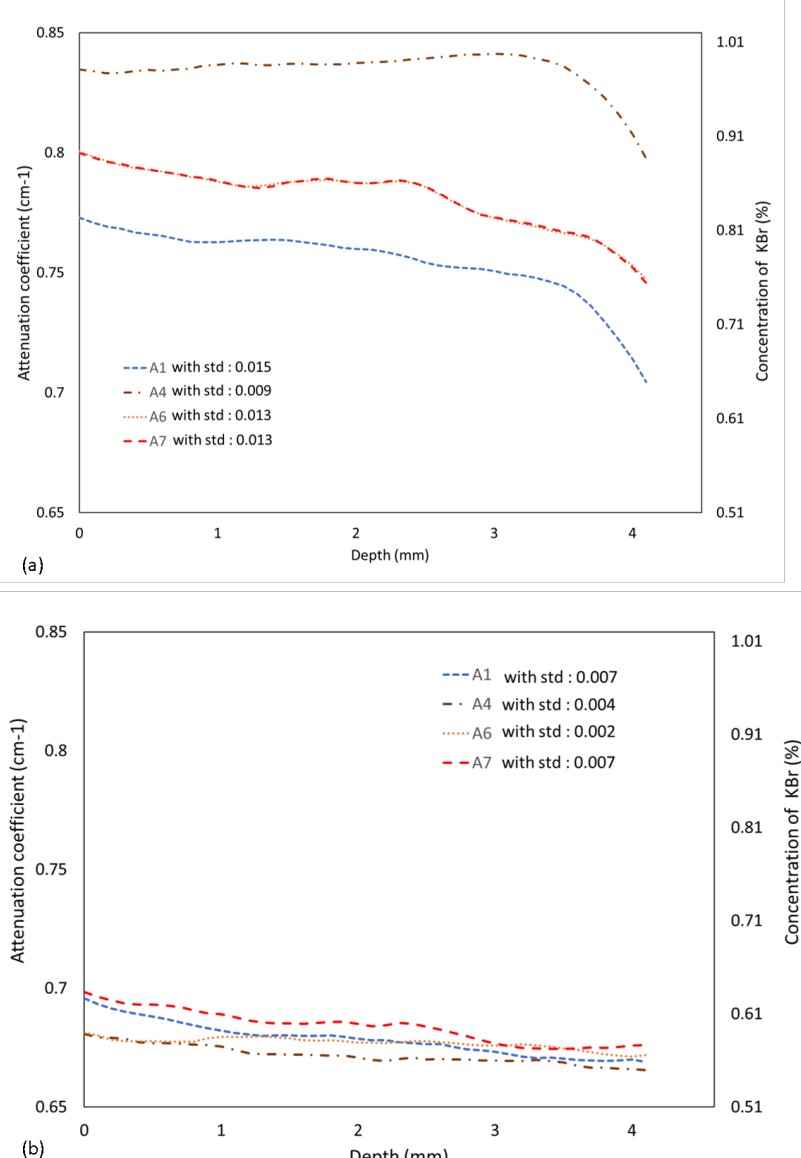

**Figure 9.** The measured attenuation coefficient in different pores as a function of depth due to PBS application on biofilms (**a**) at initial application (**b**) after 1 h of application of PBS. The legend represents the grid holes as labelled in Figure 3a. (std—standard deviation).

### 3.5. Visualization of Biofilms with Application of Contrast Agent

The X-ray µCT obtained after the PBS wash as mentioned in Section 2.10 was compared with the scan acquired before application of contrast agent to biofilms. In Figure 10 we can see the change in the attenuation coefficient before and after application of contrast agent and the difference of two images gives the region representing the contrasted biofilms attached to the grid plate. This attenuation coefficient value of the contrasted biofilms can be used to map the biofilms present in the flow cell.

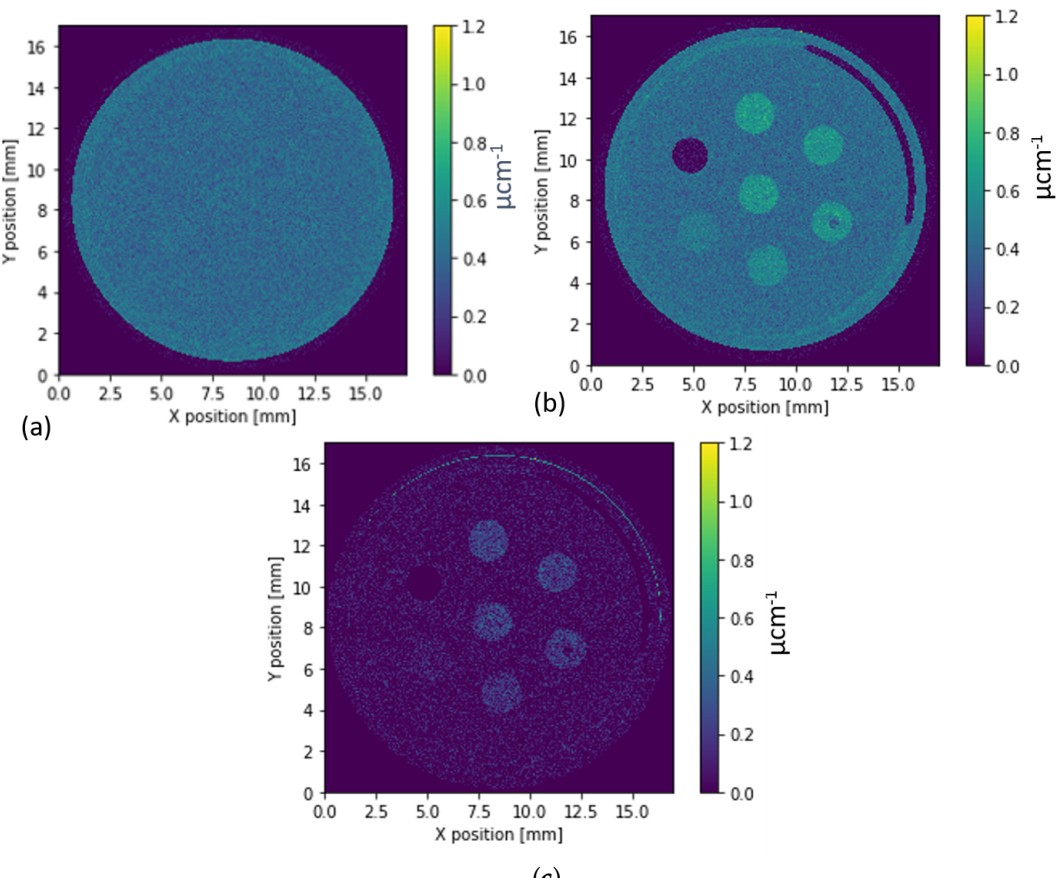

**Figure 10.** 2D cross-sectional images of the grid plate. (**a**) after TSB removal (**b**) After application of contrast agent followed by PBS wash (Section 2.10) (**c**) contrasted biofilm obtained by subtracting Figure 10a from Figure 10b.

The X-ray CT scan of the flow cell with the contrasted biofilms was normalised and in Figure 11 we can see the projections of biofilms attached to the grid plate.

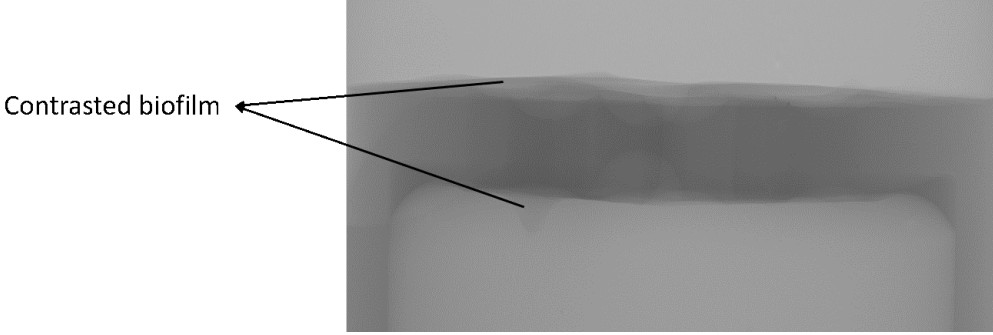

**Figure 11.** Normalised image of the porous substrate after PBS wash. The projections seen on the grid plate represent contrasted biofilms.

## 4. Discussion and Conclusions

This work focused on in-situ visualisation of biofilms inside a porous substrate using X-ray μCT. The toxic effect of different chemical compounds and the diffusion and drainage of contrast agents in biofilms were assessed. From the study, 2% $w/v$ KBr proved to be relatively less toxic as there was no reduction in biofilms population size unlike in other chemical compounds (Figure 5) and based on the contrasting power of KBr vs. its toxicity on biofilms, 5% $w/v$ KBr was used for this application.

Previously, commercially available cellulose sponges were used for studying biofilm formation in sponge pores by X-ray μCT, because of their use as a cleaning appliance both in household and industrial settings. The non-sterile conditions and humid environment at these cleaning tasks, favour the growth of bacteria in these sponges. However, the random distribution of pores with large size variability in these sponges lead to diffusion of biofilms and makes it difficult to characterize the features using X-ray μCT. The hydrophilic nature of the sponge [53] leads to absorption of contrast agents within the substrate and the homogeneous application could not be realised. The presence of dense particles in the solid matrix of the sponge [54] interferes with stained biofilms and this overlap in grey values make it difficult to segment the image. Furthermore, the partial volume effect [55] has significant contribution in substrates with large size variability. Because of these limitations, in the current study an experimental porous substrate was constructed in polymethylmethacrylate (PMMA), which acted as a model porous substrate constituting pores with a similar diameter.

The toxicity tests by plate counting for bacteria incubated in 5% *w/v* KBr solution showed a 1 $log_{10}$ reduction in biofilms population size. However, the simulated data for 5% *w/v* KBr application showed a significant distinction between the contrasted grid holes and the surrounding aqueous medium. Therefore, KBr at 5% *w/v* was applied as a contrast agent for this study to visualize biofilms in-situ using X-ray μCT based on its uptake volume in biofilms and its contrasting power. If we consider an ideal case, i.e., if all the applied contrast agent is taken up by biofilms, according to simulation, for 5% *w/v* KBr application, the contrasted biofilms in the grid hole should have an attenuation coefficient value around 2.4 $cm^{-1}$. However, in the real experiment, we have a maximum of 1.78 $cm^{-1}$ as the attenuation coefficient value of the contrasted biofilm (Figure 8b) in the grid hole. Further, if we consider the attenuation coefficient value to increase linearly with the volume of the contrast agent, the percentage of contrast agent in biofilms can be determined after subtracting the water contribution of 0.45 $cm^{-1}$. Therefore, for the attenuation coefficient value of 1.33 $cm^{-1}$, we can infer that the concentration of KBr is about 3.4% in the biofilms present in the grid hole after contrast agent application (Figure 8b). Similarly, for attenuation coefficient value of 0.39 $cm^{-1}$, the concentration of KBr is about 1% in the biofilms present in the grid hole after PBS wash (Figure 9a). These calculations give an estimate on the amount of contrast agent that is present in the biofilms.

To record the morphological changes, a confocal laser scanning microscope was used to visualize the three batches of biofilms (control, 5% *w/v* KBr solution and acute application). Although cell elongation was observed in the 5% *w/v* KBr incubated bacteria for 48 h compared to control, for 4h application of 5% *w/v* KBr no significant changes in morphology were observed. Based on the diffusion experiments, it can be assumed that 30 min application of contrast agent is sufficient for diffusion into biofilms. Considering biofilms as a porous membrane the imbibition of contrast agent solution can be explained by diffusion where the movement of fluid takes place from the higher concentration region to the lower concentration region. As biofilms are made up of around 90% of water [48] the surrounding contrast agent solution which is at higher concentration tends to diffuse into the biofilms. This diffusion is accelerated by the chemical affinity of EPS matrix towards $K^+$ ions [56,57] and $Br^-$ ions [39]. Gründling et al. [58] has elaborated on the ability of bacteria to accumulate intracellular potassium several times more than the surrounding medium. Hence the osmotic pressure along with chemical affinity towards $K^+$ ions facilitated easy passage of KBr into biofilms. However, the reason for non-uniformity in attenuation coefficient value along the depth of the grid hole could be due to the low volume of biofilms. *Pseudomonas fluorescens* are aerobic bacteria and biofilms were therefore predominantly observed at the liquid-air interface in the flow cell [59]. These biofilms were allowed to settle on the grid plate by draining out the growth medium, leading to a higher density of biofilm at the entrance of the holes than along the hole depth [60]. Therefore, on the addition of contrast agent solution, a more rapid diffusion was observed at the entrance than along hole depth. Similarly, if we consider household sponges, due to their porous open-cell structure the food soils and liquid may get impregnated randomly throughout the sponge. With sufficient aeration in the sponge, biofilms may tend to grow at the liquid-air interface in the pores of the sponge. However, since these sponges

undergo dynamic actions like squeezing and rinsing, there are considerable chances of biofilms being embedded in the pore volume. Therefore, along with the porous structure of the substrate, the forces induced by the external stimuli presumably determine the location of biofilms inside the substrate.

The diffusion experiment aimed at evaluating the diffusion of the contrast agent and PBS along with the thickness of biofilms over time. There was a sudden decrease in the attenuation coefficient value of the contrasted biofilms to almost its half when submerged with PBS solution. For segmentation, the entire grid hole was assumed to be filled with biofilms, but in reality, this is often not the case. Although enough care was taken during the drainage and application process of the fluids, some amount of biofilms present in the grid holes might have been removed. This void in the meantime gets filled with the PBS solution and hence while segmentation by thresholding based on histogram shape-based method, there is a fair chance of including the PBS for attenuation coefficient calculation.

The solubility of KBr in water is beneficial to its application on the developed biofilms and simplifies the experimental procedure. There was no precipitation of salts and a homogeneous distribution of contrast agent solution was achieved. Although both KI and KBr could be considered as contrast agents for its $K^+$ ions, the plate counting results showed that, unlike 2% $w/v$ KI, for 2% $w/v$ KBr solution there was no reduction in the bacterial population size. In addition, with the increase in the extracellular concentration of iodine its intake into the biofilms decreases [61] hence, 5% $w/v$ KI may not yield sufficient contrast enhancement. On the other hand, for 5% $w/v$ KBr there is sufficient contrast enhancement (Figure 4) with minimal effect on biofilm structure (Figure 6).

The diffusion experiments illustrated that the enhancement in the attenuation coefficient value of biofilms varies along the depth of the porous substrate. Hence, it is hard to define a single attenuation coefficient value for the entire contrasted biofilms. However, by using the average attenuation coefficient value of the attached biofilms for segmentation, the presence or absence of the biofilms at any given location in the substrate can be determined. Similarly, by subtracting the two images obtained before and after application of contrast agent an estimate on the contrasted biofilms present in the substrate can be determined.

Contrast-enhanced X-ray μCT enabled in-situ visualization of biofilms inside a porous substrate with sufficiently large contrast between the constituents of the system. Diffusion experiments further helped in understanding the imbibition process of contrast agent and enhancement in the attenuation coefficient of biofilms over time and along the depth.

**Author Contributions:** Conceptualization, M.N.B., E.R., A.S. and X.V.; methodology, A.S., M.N.B., E.R., H.S. and X.V.; software, A.S.; validation, A.S., X.V., M.N.B. and H.S.; formal analysis, A.S. and X.V.; investigation, A.S. and X.V.; resources, A.S., E.R. and X.V.; data curation, A.S. and X.V.; writing—original draft preparation, A.S. and X.V.; writing—review and editing, all authors; visualization, A.S. and X.V.; supervision, M.N.B., E.R., V.C. and H.S.; project administration, E.R., M.N.B., V.C. and H.S.; funding acquisition, E.R., V.C., M.N.B. and H.S. All authors have read and agreed to the published version of the manuscript.

**Funding:** This work is funded by the European Union's Horizon 2020 research and innovation programme under grant agreement No. 722871 in the scope of the Marie Skłodowska-Curie Action ITN BioClean. The Ghent University Special Research Fund (BOF-UGent) is acknowledged for the financial support to the Centre of Expertise UGCT (BOF.EXP.2017.0007).

**Acknowledgments:** Philippe Van Auwegem (Ghent University, Department Physics and Astronomy) is acknowledged for his assistance in building the flow cell and Daniëlle Schram (Ghent University, Department Geology) for helping in purchase and procurement of the materials. We thank Bo Gao (Ghent University, Department Physics and Astronomy) for his assistance in software installations and computer programming. David De Coster (KU Leuven, Centre of Microbial and Plant Genetics) is acknowledged for his assistance in microbiology lab.

**Conflicts of Interest:** The authors declare no conflict of interest and the funders had no role in the design of the study; in the collection, analyses, or interpretation of data; in the writing of the manuscript, or in the decision to publish the results.

**Abbreviations**

The following abbreviations are used in this manuscript:

CT      Computed Tomography
LB      Lysogeny broth
TSB     Tryptic soy broth
PBS     Phosphate buffer saline
*w/v*   weight/volume

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
