# Peer review of "Study on the Effect of Contrast Agent on Biofilms and Their Visualization in Porous Substrate Using X-ray μCT"

_applsci, doi:10.3390/app10165435_

Round 1

Reviewer 1 Report

I think the subject of this research is interesting especially at this stage.

the manuscript is well written. 

This research is focused on the contrast (kBr) which will enhance subject contrast to the level to allow a micro-CT to be able to image bacteria. 

However, this paper didn't explain why is that K and Br are used and which of the two elements is mainly enhancing the the contrast?

the above point could lead to select different kVp to optimise the contrast agent's effects

Another point regarding the scanner: is this micro-CT used in this work special? or any micro-CT can be used for such an imaging of bio-films? The authors need to explain this.

The following points needs to be addressed before this paper can be published

  • In the abstract: The last sentence; The word clearly needs to be augmented with some numbers. How clearly can you distinguish it?
  • Introduction: page 3: line 95; insert "to" between due and the
  • sentence starting line 98: This sentence needs expansion. how and why is that the attenuation of the contrast agent is higher than that of the bio-film?? please explain 
  • Figure 3-b is not clear 

Author Response

Dear reviewer, thank you for the constructive feedback which helped to improve our manuscript. Please see the attachment

Reviewer 2 Report

The manuscript addresses a visualization method to study biofilms. Biofilms may form on living or non-living surfaces and are of upmost importance in the medical, environmental and industrial settings. In the manuscript authors induce the formation of biofilms on a custom made device. These biofilms are simulated and characterized by standard methods before being observed in the Environmental Micro-CT system or EMCT Ghent University Centre for X-ray Tomography (UGCT). This reviewer recommends the publication of the manuscript with minor clarifications.

Line 77 – to be consistent name the chemicals BaSO4 and KI

Line 103 – Although references are included regarding the protocols followed, minimal description of the protocols should be mentioned since in the conclusion it is assumed that the reader knows that the biofilms were formed at liquid-air interface. Consider re-writing this session to include relevant details.   

Figure 1(b) add the grid thickness 0.5 cm

Table 1         Preculture of Bacteria at 30oC? Or 28oC? Either correct or clarify in text

Figure 5  all chemicals are easily identified except for PTA Please write out the name of the chemical even though explained in text. Add ns = no significant difference as in following figures in the manuscript

Table 2 explain what SD stands for in the legend; should be consistent with Fig 8 and 9

Figure 6 add size bar 

Figure 8, 9 std - should use same abbreviation as used in Table 2

Line 203 consider “3.4.2 Phosphate buffer saline application”, instead

Line 313 remove “s”

Line 324 consider “polymethymethacrylate” instead of PMMA helps reader

Line 331 clarify … “agent is taken up/attached to biofilms…”

Lines 356 – 359 …..” Since the biofilms grown at the liquid-air interface ( was> were) allowed to settle on the grid plate, the biofilms concentration tends to be more at the entrance of the hole than along the hole depth. Therefore on addition of contrast agent solution the rapid diffusion is observed at the entrance than along its depth.” Re-write to clarify ….

Question regarding discussion/conclusion presented Lines 354 – 359:  Does it mean that biofilms grow at the liquid/air interface? How does this relate to natural environments? For example let’s consider a household sponge, should we expect to observe biofilm coverage of the pores to be more significant close to the outside surface of the sponge compared to the core volume?        

Author Response

Dear reviewer, thank you for the constructive feedback which helped to improve our manuscript. Please see the attachment.

Reviewer 3 Report

The work investigates the effect of KBr and other contrast agents on biofilms and its visualization in porous substrate using X-ray μCT. This manuscript addresses an area of interest and is clearly written. I have some minor comments that could improve the study.

Title: visualization

Line 9 in situ in cursive

Line 17 bactericidal

Add in the Abstract the bacteria that was used

Line 29 decreased

Line 29 mention what gradients, pH, oxygen, nutrients…

Line 43 stress conditions

Line 47 allow

Line 48 implements?

Line 54 imaging porous materials

Line 68 remove fluid

Line 81-82 et al in cursive

Line 93 Alginic acid? Is that alginate?

Line 95 due to the

I wouldn’t review other studies in the introduction. I would move this section to the discussion.

Line 101 mature?

Line 106 cultures

Line 113 real world?

Line 114 polymethylmethacrylate

Figures 1 and 2 can be merged. Also, the pores in figure 2 seem very different from figure1b.

Line 140 and 141 in situ in cursive

Line 147 in-house

Line 190 approximately

Line 194 and table 1 culture

Table 1 you mention sections 2.8, 2.9 and 2.10 before it appears in the text, I would move the table to after those sections are described. Add how long was the 5% KBr incubation (4h, 1h, 30 min?)

Line 207, if I understand correctly table 1, you also performd a PBS wash of the biofilm to remove planktonic bacteria, not just to collect the biofilm. Please add this information in this section as well.

Line 266 is that p=0.0023 for the 2% of FeSO4 or Zn SO4? What is the p for the other contrast agent? If it is the same value, it is more comprehensive to write it twice.

Figure 5 and 7 y axis, write that it is log 10. Add also in the material and methods the statistical test that was performed. What post-hoc test was used? Tukey…?

Table 2 the total biomass seems to be reduced in the 5% KBr case, from 0.13 to 0.10. If 5% KBr kills 1 log bacteria as you have shown in figure 5, I wouldn’t interprete the biomass/cell as an increase in biomass production but as a decrease in the number of cells. Is the total biomass value an OD read? If not what is it? What units does it have?

Figure 6, please add scale bar

Figure 7.- colony forming units per cm2, not of biofilm

Figure 8 and 9.- at what time point is the std calculated?

Line 294 on the contrary

Line 308, 330, 381 italianize in situ

Line 313 remove s

Line 343 explain which one is the acute application, 4h or 48h? add the incubation time with KBr in the following sentence, it is a little bit unclear which is which as it is currently described.

Line 371 Actually, a 5% KBr concentration is more bactericidal than 2%KI, figure 5. According to table 1 and if I have understood correctly 5% is the KBr concentration used in the experiments. Specify that in this sentence you are referring to 2% KBr and justify why you have used 5% KBr for the experiments.

Line 375 Hence,

Author Response

(The authors gave the same response as above.)
